# In Vitro Identification of Phosphorylation Sites on TcPolβ by Protein Kinases TcCK1, TcCK2, TcAUK1, and TcPKC1 and Effect of Phorbol Ester on Activation by TcPKC of TcPolβ in *Trypanosoma cruzi* Epimastigotes

**DOI:** 10.3390/microorganisms12050907

**Published:** 2024-04-30

**Authors:** Edio Maldonado, Paz Canobra, Matías Oyarce, Fabiola Urbina, Vicente J. Miralles, Julio C. Tapia, Christian Castillo, Aldo Solari

**Affiliations:** 1Programa de Biología Celular y Molecular, Instituto de Ciencias Biomédicas (ICBM), Facultad de Medicina, Universidad de Chile, Santiago 8380453, Chile; paz.canobra@ug.uchile.cl (P.C.); matias.oyarce@ug.uchile.cl (M.O.); fabi.urbina1516@gmail.com (F.U.); jtapiapineda@uchile.cl (J.C.T.); 2Departamento de Bioquímica y Biología Molecular, Universidad de Valencia, 46110 Valencia, Spain; vicente.j.miralles@uv.es; 3Programa de Anatomía y Biología del Desarrollo, Instituto de Ciencias Biomédicas (ICBM), Facultad de Medicina, Universidad de Chile, Santiago 8380453, Chile; ccastillor@uchile.cl

**Keywords:** DNA pol β, protein kinases, phosphorylation, signal transduction, *T. cruzi*

## Abstract

Chagas disease is caused by the single-flagellated protozoan *Trypanosoma cruzi*, which affects several million people worldwide. Understanding the signal transduction pathways involved in this parasite’s growth, adaptation, and differentiation is crucial. Understanding the basic mechanisms of signal transduction in *T. cruzi* could help to develop new drugs to treat the disease caused by these protozoa. In the present work, we have demonstrated that Fetal Calf Serum (FCS) can quickly increase the levels of both phosphorylated and unphosphorylated forms of *T. cruzi* DNA polymerase beta (TcPolβ) in tissue-cultured trypomastigotes. The in vitro phosphorylation sites on TcPolβ by protein kinases TcCK1, TcCK2, TcAUK1, and TcPKC1 have been identified by Mass Spectrometry (MS) analysis and with antibodies against phosphor Ser-Thr-Tyr. MS analysis indicated that these protein kinases can phosphorylate Ser and Thr residues on several sites on TcPolβ. Unexpectedly, it was found that TcCK1 and TcPKC1 can phosphorylate a different Tyr residue on TcPolβ. By using a specific anti-phosphor Tyr monoclonal antibody, it was determined that TcCK1 can be in vitro autophosphorylated on Tyr residues. In vitro and in vivo studies showed that phorbol 12-myristate 13-acetate (PMA) can activate the PKC to stimulate the TcPolβ phosphorylation and enzymatic activity in *T. cruzi* epimastigotes.

## 1. Introduction

Chagas disease, or American trypanosomiasis, is caused by *Trypanosoma cruzi*, a group of single-flagellated parasites classified in discrete typing units (DTUs), which are found in a wide range of geographic areas [1,2,3]. This disease has two phases: the initial acute phase, which can last several weeks, and the chronic phase, which might continue for decades after the initial infection and can generate mega-organs syndrome, cardiomyopathy, and sudden death [1]. Interestingly, about 2/3 of infected persons never develop the characteristics and symptoms of the disease, suggesting that host genetic factors are associated with mitochondrial dysfunction and inflammation in cardiomyopathies of Chagas disease [4]. In their life cycle, the parasite *T. cruzi* circulates in two types of hosts (vertebrate and invertebrate) and is transmitted to vertebrate hosts by stercorary triatomine insects distributed in America [5]. However, due to human migrations, Chagas disease has spread worldwide and can be transmitted horizontally during pregnancy. Currently, Chagas disease cases are found in many countries outside the American continent [6,7].

*T. cruzi* belongs to the Excavata supergroup of Protist, typified with unique discoidal cristae mitochondria present in subgroups such as euglenoids and Kinetoplastids [8]. The kinetoplast DNA is a network of circular DNAs consisting of thousands of interlocked DNA circles of two types: thousands of minicircles and dozens of maxicircles corresponding to the mitochondrial DNA. A representation of the kinetoplast domains and a proposed molecular mechanism of minicircle replication can be found elsewhere [9]. *T. cruzi*, like all trypanosomatids, can efficiently respond to quick environmental changes, though they cannot regulate gene expression at the transcriptional level as higher eukaryotes do. They can mainly regulate responses at the post-transcription or post-translational levels [10]. Trypanosomatids respond to extracellular and intracellular signals as they adapt quickly to new environments inside their various hosts. Adaptations to different environments with different nutrient availability, temperatures, and immune responses require significant changes in the control and regulation of gene expression. During these parasite–environment interactions, several signaling molecules and pathways can be activated [11]. Protein phosphorylation is a reversible post-translational modification that constitutes a key mechanism to control protein function via activation/inactivation and/or changes in subcellular localization.

Several proliferative forms of *T. cruzi* in the life cycle are non-infective (epimastigotes in the last portion of the triatomine intestine and amastigotes inside the parasitophorous (vacuole in vertebrate cells) which proliferate in these environments. Meanwhile, other parasitic forms, such as metacyclic trypomastigotes, are found in triatomines and bloodstream trypomastigotes in vertebrate hosts. Both are non-proliferative and differentiate from proliferative forms triggered by extracellular signals to quickly remodel their whole structure and surface antigens, controlling degradation and protein synthesis to gain cellular infectivity [12,13]. Alternatively, metacyclic trypomastigotes can differentiate into amastigotes in rich nutrient and acidic media [14].

The proliferation of *T. cruzi* epimastigotes utilizes closed mitosis, a process where the nuclear membrane remains intact during cell division. Several regulated proteins and phosphoproteins playing essential roles in the control of cell division of *T. cruzi* epimastigotes have been described [15]. *T. cruzi* proliferation involves the replication of mitochondrial and nuclear genomes and DNA repair of mutations due to external and internal chemical and physical mutagenic agents that damage DNA. The mitochondrial genome accumulates mutations due to the reactive oxygen species (ROS) produced by electron leakage during oxidative phosphorylation and the reactive nitrogen species (RNS) produced in *T. cruzi*-infected macrophages. DNA replication involves a considerable number of accessory proteins and enzyme functions. Among them are the DNA polymerases and accessory factors, which are extensively studied [16,17,18,19,20]. Several DNA polymerases are found in mammalian cells, namely Pol alpha (Pol α), Pol beta (Pol β), Pol gamma (Pol γ), and Pol epsilon (Pol €), among others. DNA Pol α plays a crucial role in nuclear DNA replication, DNA Pol β plays a role in nuclear DNA repair, and DNA Pol γ is involved in mitochondrial DNA replication and repair [16,17,18,19,20].

DNA polymerases play a key role in DNA replication and cell division. Earlier biochemical studies in *T. cruzi* epimastigotes have led to the characterization of different DNA polymerase fractions with different properties to those found in mammalian cells [21,22,23]. Interestingly, some of these were similar to other DNA polymerase activities found in other trypanosomatids and in the algae *Chlorella* [24,25,26,27]. Later, a third and major DNA polymerase activity was described in epimastigotes, which was different from those described earlier [28]. This DNA polymerase activity was purified to near homogeneity and consists of two polypeptides with a molecular mass of 50 and 55 kDa. Only the 55 kDa polypeptide presented in gel enzymatic activity [29]. A similar DNA Pol β was purified from the trypanosomatid *Crithidia fasciculata* [30,31]. The cloning of the *T. cruzi* DNA Pol β gene from the *T. cruzi* Miranda clone (Tc I DTU) revealed the protein sequence and a size of 403 amino acids, including the mitochondrial targeting sequence [32]. The preliminary analysis of the DNA Pol β amino acid sequence showed four putative phosphorylation sites according to the specificity of mammalian protein kinases: three for casein kinase 2 (CK2) and one for protein kinase C (PKC) [32]. Interestingly, these sites are located in a protein segment without any similarity between the *T. cruzi* enzyme and the mammalian DNA Pol β and correspond to an extra segment between the amino acids 316 and 373, suggesting this might be a potential domain to regulate enzyme activity. Further studies with *T. cruzi* DNA Pol β (named TcPolβ) confirmed that two enzyme forms exist (molecular masses of 50 and 55 KDa) in epimastigotes and trypomastigote forms. These forms are named L and H, and only the one with higher molecular mass (H form) is phosphorylated and active in DNA synthesis [33]. The expression and purification of a recombinant TcPolβ have also been reported [33,34].

Mammalian DNA Pol β is nuclear located, and it is involved in base excision repair (BER), while IN *T. cruzi* localizes to the kinetoplast and participates in kinetoplast DNA repair and replication. It was described that the overexpression of TcPolβ in transfected *T. cruzi* makes cells more resistant to hydrogen peroxide [35]. Later, the effect of oxidative stress on the expression and function of TcPolβ on epimastigotes and trypomastigote forms exposed to hydrogen peroxide treatment was reported. The phosphorylated TcPolβ protein levels can increase in response to hydrogen peroxide damage in both parasite forms [34].

Since TcPolβ plays a key role in kinetoplast DNA repair and replication, this enzyme’s expression and enzymatic activity must be tightly regulated. To understand the molecular mechanisms underlying this regulation, we defined three research goals for this work: first, the effect of growth signals on the expression and phosphorylation of TcPolβ; second, the identification of the in vitro phosphorylation sites on TcPolβ by protein kinases TcCK1, TcAUK1, TcCK2, and TcPKC1; and third, the effect of PMA on the phosphorylation of TcPolβ by TcPKC. In this work, we reported the effect of growth signals on the expression of TcPolβ. We also identified the in vitro phosphorylation sites on this DNA polymerase by TcCK1, TcAUK1, TcCK2, and TcPKC1 by MS and confirmed these modifications by using anti-phosphor antibodies. Furthermore, we demonstrated that PMA could activate TcPKC1 and consequently increase the phosphorylation and enzyme activity of TcPolβ in vitro. PMA can also in vivo activate TcPKC to phosphorylate TcPolβ in *T. cruzi* epimastigotes.

## 2. Materials and Methods

Treatment of *T. cruzi* epimastigotes and trypomastigotes with heat-inactivated FCS, and PMA.

Epimastigotes (Y strain) were cultured in liver infusion tryptose serum medium (LIT) at 28 °C, supplemented with 10% heat-inactivated FCS (Capricorn Scientific, Ebsdorfergrund, Germany) [36] until reaching mid-log phase and then treated with PMA at 2 µM final concentration.

Trypomastigotes (Y strain) were obtained from the infection of semi-confluent Vero cells (ATCC CCL-81) with trypomastigotes from a previous culture for 72 h in a cell incubator at 37 °C and 5% CO_2_ [37]. Trypomastigotes invaded cells, replicated intracellularly as amastigotes, lysed the cells, and were collected from the supernatant through low-speed centrifugation (500× *g*) to remove mammalian cell debris. The parasites were then transferred to a new tube and centrifuged at 6000× *g* for sedimentation. The number of parasites was determined using a Neubauer chamber and subsequently suspended in RPMI1640 or the media supplemented with 5% heat-inactivated FCS.

### 2.1. Protein Expression and Purification

Encoding genes for TcCK1, TcCK2, TcAUK1, TcPKC1, and TcPolβ were inserted in pET15b (Novagen, Merck KGaA, Darmstadt, Germany) and transformed in BL21 (DE3) cells, and their protein expression was induced by 0.5 mM IPTG addition [38,39]. Recombinant TcCK1, TcCK2 (α–β subunits), TcAUK1, TcPKC1, and TcPolβ proteins were purified according to references [38,39] using NTA-Ni-agarose resin. Protein concentration was determined by the Bio-Rad protein assay Kit (Hercules, CA, USA) using BSA as standard. The purity of the recombinant polypeptides was assayed by PAGE-SDS, followed by colloidal Coomassie Blue G-250 staining (Invitrogen, ThermoFisher Scientific, Waltham, MA, USA). Colloidal Coomassie Blue G-250 can detect less than 10 ng of protein on a PAGE-SDS following the manufacturer’s instructions.

### 2.2. Phosphorylation Assays

Phosphorylation assays were performed as described in the references [38,39], including 1 mM of ATP (Promega Corporation, Madison, WI, USA). Amounts and combinations of the different recombinant proteins were added as described in each Figure legend.

### 2.3. DNA Polymerase Activity Gel

The TcPolβ activity was detected in situ in a PAGE-SDS gel as described in the literature [40] with major modifications. Reactions containing different amounts of TcPolβ (20 µL final volume) were mixed with 5 µL of 5× Laemmli buffer (250 mM Tris, pH 6.8, 10% *w*/*v* SDS 0.1% Bromophenol Blue and 50% glycerol) supplemented with 5 mM DTT and loaded in a 9 % PAGE-SDS gel containing 100 µg per each ml of gel solution of activated calf thymus DNA. When the run was complete, the gel was washed and TcPolβ was renatured in the folding buffer as described in the references [38,40]. After this step, the gel was incubated in folding buffer supplemented with 20 µM of each dATP, dCTP, dGTP, and 0.5 µM of Biotin-11-dUTP (Jena Bioscience, Jena, Germany) plus 0.1 µM of dTTP. Incubation was performed for 20 h at room temperature with gentle agitation. Then, the gel was washed 4 times for 15 min (each wash) in 50 mM Tris-HCl, 150 mM NaCl, pH 8.0, and incubated with a 1/2.000 dilution of Streptavidin-Alkaline Phosphatase (AP) conjugated (Promega Corp, Madison, WI, USA) in washing buffer. Incubation was performed for 2 h at room temperature. After this step, the gel was washed, as described above, and developed overnight with 100 µg/mL NBT and 60 µg/mL BCIP (Bio-Rad, Hercules, CA, USA) in 100 mM Tris-HCl, 100 mM NaCl, and 12 mM MgCl2, pH 9.5. The gels were recorded, and the signals were quantified using the ImageJ software.

### 2.4. Protein Phosphorylation Analysis

Twenty micrograms (20 µg, approximately 500 pmol) of TcPolβ were phosphorylated with 100 pmol of each protein kinase (TcCK1, TcCK2, TcAUK1, and TcPKC1) in 200 µL reaction volume with 1 mM ATP as described above. Phosphorylated TcPolβ was separated in a 9% PAGE-SDS gel and visualized by colloidal Coomassie Blue G-250 staining. The band corresponding to the TcPolβ polypeptide was cut out from the gel and sent for phosphor peptide and phosphorylation site analysis at Creative Proteomics (Shirley, NY, USA). Briefly, the phosphorylated TcPolβ in the gel slices was treated with trypsin, and then the peptides were recovered from the gel slices. The peptides were separated by NanoLC and identified by MS/MS scan. The raw MS files were analyzed and searched against the RNC61524.1 (TcPolβ) protein reference sequence using the Maxquant (1.6.2.6) program. The detailed identified peptides and phosphorylation sites were listed in an Excel sheet and then located in the TcPolβ reference sequence (see Section 3).

### 2.5. Western Blot Analysis

TcPolβ was phosphorylated as described above in a 20 µL final volume. Phosphorylated TcPolβ was separated in a 9% PAGE-SDS gel and transferred to Immobilon-E membranes, which were blocked in bovine serum albumin (BSA) at 4% *w*/*v* in TTBS (50 mM Tris-HCl, 150 mM NaCl, 0.1% *v*/*v* Tween 20, pH 8.0) for 2 h at room temperature. After, the membranes were incubated overnight with anti-phosphor Ser-Thr-Tyr from Abcam (Catalog ab15556, Cambridge, UK) at a dilution of 1/300 in 4% *w*/*v* of BSA in TTBS buffer or anti-phosphor Tyr from Cell Signaling Technologies (Catalog 9411, Danvers, MA, USA) at a dilution of 1/1.000 in 4% *w*/*v* of BSA in TTBS at 4 °C with gentle rocking. After this step, the membranes were washed with TTBS and incubated with anti-mouse IgG conjugated to horseradish peroxidase (HRP) at a dilution of 1/10.000 (Cell Signaling Technologies, Catalog 7076, Danvers, MA, USA) in TTBS for 45 min at room temperature. Once the incubation was complete, the membranes were washed with TTBS, overlayed with ECL Ultra Western HRP substrate from Merck KGaA (Catalog WBULS0100, Darmstadt, Germany), and exposed to an X-ray film for 20 min. The films were developed, dried, and scanned. The Western blot for TcPolβ was performed as described in the reference [33]. Images were quantified using the ImageJ 1.52a software (imagej.net).

### 2.6. Statistical Analysis

Results were expressed as the mean *±* standard deviation (SD). Experiments were repeated at least three times. Statistical analyses were performed using the ImageJ 1.52a software for area and SD calculations. Graphics were constructed by using Microsoft Excel 2019 software based on the calculations of the previous value by the ImageJ 1.52a software.

## 3. Results

### 3.1. Fetal Calf Serum Increases Tcpolβ Protein Synthesis and Phosphorylation in Trypomastigotes

Since Fetal Calf Serum (FCS) contains different growth factors, attachment factors, vitamins, minerals, metabolites, and lipids, we compared the expression levels of TcPolβ in trypomastigotes incubated in RPMI1640 medium and RPMI1640 medium supplemented with 5% *v*/*v* of heat-inactivated FCS. In Figure 1, it can be observed that the levels of TcPolβ in cell-cultured trypomastigotes incubated in RPMI1640 medium supplemented with 5% *v*/*v* of heat-inactivated FCS are higher than those of parasites incubated in RPMI1640 without FCS (compare lanes 1–3 with lanes 4–8). Figure 1 shows that both forms, H and L, are augmented in trypomastigotes incubated with heat-inactivated FCS compared to those incubated in media alone. The results presented in Figure 1 are not due to differences in the number of cells since both cultures contain the same number of trypomastigotes (250.000 cells). Moreover, the levels of a constitutive protein, such as tubulin, do not change in the cells incubated in media alone or media supplemented with heat-inactivated FCS (Figure 1, lower panel). These results indicate that FCS components signal TcPolβ protein synthesis and phosphorylation in trypomastigotes since the antibody is able to recognize both forms of TcPolβ (H and L), and both forms are augmented.

### 3.2. Tcpolβ DNA Synthesis Activity Can Be Detected in the Gel by a Colorimetric Assay

To study the effects of phosphorylation of protein kinases on the activity of the recombinant TcPolβ, we developed a novel DNA polymerase activity gel assay followed by colorimetric detection. Figure 2a shows a dose–response curve of recombinant TcPolβ in the DNA polymerase activity gel assay. It is observed that TcPolβ (50 kDa) produces a linear dose–response in a range of concentrations between 40–640 ng (Figure 2b). The optimal minimal amount of TcPolβ detected with this assay is 40 ng; however, this is a high amount compared with the DNA polymerase Klenow large fragment (80 kDa MW), which has a detection limit of 0.5 ng in this assay (see Appendix A). Comparing the activity of the DNA polymerase Klenow large fragment with the unphosphorylated TcPolβ, we conclude that the DNA polymerase Klenow large fragment is approximately 160-fold more active in this particular assay. Probably, the unphosphorylated TcPolβ is not very active in vitro, due to a lack of this post-translational modification. Figure 2b shows the signal quantification of Figure 2a and it can be observed that a linear dose–response is produced in a range of concentration of 40–640 ng of TcPolβ.

### 3.3. In Vitro Determination of the Phosphorylation Sites on TcPolβ by TcCK1, TcCK2, TcAUK1, and TcPKC1

Earlier, we demonstrated that TcPolβ is an in vitro substrate of several protein kinases, including TcCK1, TcCK2, TcAUK1, TcPKC1, TcPKC2, and Wee1-like [38,39]. However, the exact phosphorylated residues from the different protein kinases are unknown. To determine the exact residues phosphorylated by some of these protein kinases, we phosphorylated in vitro recombinant TcPolβ (50 kDa) by TcCK1 (50 kDa), TcCK2 holoenzyme (*α* 40 kDa; β 48 kDa), TcAUK1 (55 kDa), and TcPKC1 (120 kDa). All proteins used in this study were at least 90% pure as judged by PAGE-SDS, followed by colloidal Coomassie Blue G-250 staining (Figure 3a). After phosphorylation by the different protein kinases, the phosphorylated TcPolβ was separated on a PAGE-SDS gel, digested by trypsin, and subjected to MS detection of the phosphor-peptides (see Appendix A). These results are presented in Table 1. Several residues were found to be phosphorylated by the different protein kinases. TcCK1 phosphorylates 12 residues of Thr and Ser, whereas TcCK2 and TcAUK1 phosphorylates 3 residues of Thr and Ser. TcPKC1 can phosphorylate 7 residues of Thr and Ser. Unexpectedly, protein kinases TcCK1 and TcPKC1 can phosphorylate one Tyr residue each at the C-terminal region of TcPolβ. These phosphorylated Tyr residues are different for TcCK1 and TcPKC1. Interestingly, most of the phosphorylated Ser and Thr residues on TcPolβ are identical for the four protein kinases (see Section 4).

### 3.4. Tcpolβ Phosphorylation Is Detected by Western Blot Analysis with Specific Anti-Phosphor Antibodies

We further confirmed the phosphorylation of TcPolβ by protein kinases TcCK1, TcCK2, TcAUK1, and TcPKC1 by Western blot analysis using antibodies against phosphor Ser-Thr-Tyr. These results are presented in Figure 4. These antibodies can recognize phosphorylated TcPolβ by TcCK1 in a specific fashion (Figure 4a, lanes 1–3) since they do not react with unphosphorylated TcPolβ or TcCK1 (lanes 4 and 5, respectively). Protein kinase AURK can also phosphorylate TcPolβ in a specific manner (Figure 4b, lanes 1–3) since the antibodies do not recognize unphosphorylated TcPolβ or AURK (lanes 4 and 5, respectively). On the other hand, TcPKC1 can phosphorylate TcPolβ (Figure 4c, lanes 1–3). The antibodies do not react with unphosphorylated TcPolβ or TcPKC1 [see Figure 4c, lanes 4 and 5, respectively]. We could not detect TcPolβ phosphorylation by TcCK2, probably due to the low phosphorylation levels by this protein kinase. These results indicate that TcCK1, TcAUK1, and TcPKC1 can in vitro phosphorylate TcPolβ, and this event can be detected with anti-phosphor antibodies.

We also investigated whether anti-phosphor Tyr antibodies, which exclusively recognize phosphor Tyr residues, could detect this modification on TcPolβ since the MS data indicated that TcPolβ could be phosphorylated at Tyr residues by TcPKC1 and TcCK1. Using Western blot analysis, we could not detect the phosphorylation of TcPolβ by TcCK1 or TcPKC1. This is most likely due to the low phosphorylation levels on TcPolβ, since only one Tyr residue is phosphorylated as the MS data indicate. However, we detected strong autophosphorylation of TcCK1 (Figure 4d, lanes 1–3). The autophosphorylation is specific to TcCK1 since, in the absence of ATP, we did not detect a signal (Figure 4d, lanes 4–6). On the other hand, we did not detect the autophosphorylation of TcPKC1 on Tyr residues by Western blot analysis. The results demonstrated that TcCK1 can phosphorylate Tyr residues on TcPolβ, as detected by MS, and on TcCK1 itself as revealed by Western blot analysis.

### 3.5. Phosphatidylserine and PMA Can Directly Stimulate TcPKC1

Since TcPolβ phosphorylation can be detected by Western blot with anti-phosphor antibodies, we next investigated, using this method, whether PMA and phosphatidylserine (PS) could stimulate TcPolβ phosphorylation by TcPKC1. Figure 5 shows the results of these experiments. We limited the substrate (TcPolβ) to perform these experiments to see an optimal stimulation signal. It can be observed that PS stimulates TcPolβ phosphorylation at 60 µg/mL as the optimal concentration (Figure 5a, lane 3). Higher concentrations of PS (120 and 240 µg/mL, lanes 4 and 5, respectively) do not further stimulate TcPolβ phosphorylation. Also, PMA stimulates the phosphorylation activity of TcPKC1 (Figure 5b, lanes 2–6). Maximal stimulation of PKC1 by PMA is around 1–2 µM. These results indicate that PKC1 can be activated by PS and PMA, probably owing to the PhoX domain present at the N-terminus of this protein kinase or to another additional domain present in the polypeptide.

### 3.6. PMA Activates TcPKC1 to Phosphorylate TcPolβ and Increases Its DNA Polymerase Activity

Next, we investigated whether TcPolβ could potentiate its DNA synthesis activity by TcPKC1 phosphorylation. We used the novel DNA polymerase gel activity assay developed in Figure 2 to investigate this. The results are shown in Figure 6, and it is seen that TcPolβ, in the presence of ATP and TcPKC1, can augment its DNA synthesis activity (Figure 6a, compare lanes 1 and 2). This is the effect of phosphorylation on the DNA polymerase since TcPKC1, in the absence of ATP, does not stimulate the activity of TcPolβ (Figure 6a, lane 4). Also, the effect of PMA, a specific PKC activator, was investigated by using this assay. Figure 6 shows that PMA strongly stimulates the DNA synthesis activity of TcPolβ (Figure 6a, lanes 5–8). The stimulation is due to the effect of PMA on TcPKC1 since DMSO, the PMA solvent, does not further stimulate the PKC phosphorylation activity on TcPolβ and the consequent increase in DNA synthesis (Figure 6a, lane 3). These results indicate that PKC1 phosphorylation of TcPolβ can stimulate its DNA synthesis activity, and PMA can also stimulate the phosphorylation activity of TcPKC1 as it does in higher eukaryotes. The activity of TcPKC1 in the presence of PMA is stimulated about 5-fold, with 1.0 µg/mL, as can be observed in Figure 6b.

### 3.7. PMA can In Vivo Activate PKC to Phosphorylate TcPolβ in Epimastigotes

Since PMA could in vitro activate TcPKC1 at a concentration of 2 µM (see Figure 6a), we investigated whether TcPolβ phosphorylation could be increased in vivo in *T. cruzi* epimastigotes and trypomastigotes cells. We treated epimastigotes and trypomastigotes with 2 µM of PMA to perform these experiments. We measured the levels of TcPolβ H and L forms at 0, 3, 6, and 12 h after treatment by Western blot using anti-TcPolβ antibodies. After 12 h of PMA treatment, the parasites remained viable and mobile. In epimastigotes, the phosphorylated H form of TcPolβ was found to be augmented compared with the control (DMSO treated), as seen in Figure 7a. At 3 h post-treatment with PMA, there is an increase in the H form, and it was maintained until 12 h after the treatment (Figure 7a, lanes 6–8) compared with the control (lanes 2–4). The tubulin control (lower panel, lanes 1–8) does not vary significantly. Trypomastigotes did not respond to the treatment with PMA under the same tested conditions (2 µM) as the epimastigotes did. These results indicate that PMA can in vivo stimulate TcPKC enzymes to increase the phosphorylation levels of TcPolβ in epimastigotes. Quantification of the signals on the Western blot by using the Image J program confirms that there is an increase in the H form in epimastigotes treated with PMA (Figure 7b, lanes 5–8) compared with DMSO-treated epimastigotes (Figure 7b, lanes 2–4). The increase is about 2–3-fold from 3 h post-treatment. The L form of TcPolβ tends to augment equally in both groups of parasites (Figure 7c, lanes 1–8). The tubulin control does not change significantly, whether in DMSO- or PMA-treated epimastigotes (Figure 7d).

## 4. Discussion

Signal transduction is essential in regulating important functions in unicellular and multicellular organisms since it can convert an extracellular signal into specific cellular responses. Mostly, signal transduction starts with a signal to a membrane-bound receptor and ends with a change in cell function. The signal can produce changes in the cell, either in gene expression in the nucleus or in the activity of cytoplasmic enzymes. Signal transduction very often alters the phosphorylation status of key target proteins, therefore altering the function of these proteins. Intracellular signal transduction pathways are most of the time activated by second messenger molecules, including cAMP, cGMP, Ca^+2^, nitric oxide, and lipophilic second messenger molecules such as DAG, ceramide, and phosphatidylserine (PS), among others. These second messenger molecules can activate protein kinases to phosphorylate vital target proteins to activate or inhibit their activity. In eukaryotes, protein kinases are ubiquitous and play a role in many different intracellular signaling pathways affecting processes such as differentiation, cell growth, proliferation, cell motility, and apoptosis, among others [41].

*T. cruzi* has a complex life cycle involving four morphogenic stages in different hosts [42], and the parasite must adapt to hostile environments to survive, such as starvation and host responses, by differentiation to a specific life form. In this parasite, the second messenger system can regulate cell growth and differentiation by modulating protein phosphorylation of a set of protein substrates essential for these processes [11]. Several protein kinases play key roles in trypanosomatid protozoa, and disrupting their activity harms the parasite [41,43]. It is well known that reversible protein phosphorylation is one of the most important mechanisms for quickly regulating adaptive responses to intra- and extracellular signals in several organisms [44]. Characterization of the PKC activities in *T. cruzi* epimastigotes demonstrated several PKC isoforms, which were stimulated by DAG and PMA [45]. Biochemical and immunological studies have revealed the existence of several PKC isoforms as detected by Western blot analysis [46]. Also, a soluble form containing PKC autophosphorylation activity was characterized using immunoaffinity chromatography with a heterologous antibody [45]. Several PKC activities have also been described in the bloodstream and procyclic forms of *T. brucei* using DEAE-cellulose columns and a monoclonal antibody raised against mammalian PKC [47]. It was determined that oleic acid triggers the differentiation of *T. cruzi* epimastigotes into metacyclic trypomastigote forms through a cell signaling pathway involving DAG and PKC activation. These observations provided the first piece of evidence that PKC has a biological role in metacyclogenesis in *T. cruzi.* Also, it has been demonstrated that oleic acid can trigger a Ca^+2^ signal in epimastigotes, which is necessary for metacyclogenesis [48].

In this study, we have found that FCS increased in less than 30 min, the levels of both forms of TcPolβ in tissue-cultured trypomastigotes but not in the replicative epimastigote form, indicating that FCS provides a signal to stimulate both protein synthesis and phosphorylation of this DNA polymerase. This result suggests that trypomastigotes have the target receptor that recognizes an FCS factor that activates the synthesis and phosphorylation of TcPolβ. FCS provides growth factors and hormones, binding and transport proteins, attachment and spreading factors, additional amino acids, vitamins, and trace elements; fatty acids and lipids; protease inhibitors; and detoxification (due to binding and activation) [49]. A growth factor and/or fatty acids and lipids might provide a cell signal to quickly activate protein synthesis and a protein kinase, which can phosphorylate TcPolβ in trypomastigotes. This indicates that TcPolβ levels and the activity are regulated and modulated by signal transduction pathways when (non-proliferative) trypomastigote forms adapt to a new environment.

In earlier work, we reported that TcPolβ is in vitro phosphorylated by TcCK1, TcCK2, TcAUK1, and TcPKC1 and 2 [38,39]. Using recombinant TcPolβ as the substrate and different purified recombinant protein kinases from *T. cruzi*, the question of whether the in vitro phosphorylation of TcPolβ modifies its enzyme activity was assessed. It was found that the in vitro phosphorylation of TcPolβ augments its activity by the phosphorylation of recombinant TcCK1, TcCK2, and TcAUK1 [38]. Also, in vivo studies of phosphorylation of *T. cruzi* TcPolβ were performed [39]. Four amino acid residues were in vivo identified as modified by phosphorylation in *T. cruzi* epimastigotes TcPolβ. Two are phosphorylated in Thr, one in Ser, and one in Tyr. The in vivo phosphorylated residue is Tyr35, which is present in the conserved lyase domain of TcPolβ, while Thr123, Thr137, and Ser286 are located in the catalytic DNA polymerase domain. Based on the higher eukaryote consensus phosphorylation sequences, it was assigned that TcCK2 phosphorylates Thr123 and Thr137, while protein kinase C-like enzymes phosphorylates Ser286 [39]. Currently, the protein kinase that in vivo phosphorylates Tyr35 is unknown.

Also, we reported that TcPolβ is in vivo phosphorylated by TcCK2 and TcPKC, based on the consensus phosphorylation sites of higher eukaryotic protein kinases [39]. This report analyzes the in vitro phosphorylation sites on TcPolβ by TcCK1, TcCK2, TcAUK1, and TcPKC1 using MS analysis. We found that effectively, these protein kinases can in vitro phosphorylate TcPolβ on multiple Ser and Thr residues and also that TcCK1 and TcPKC1 can in vitro phosphorylate a Tyr residue on TcPolβ. The last result was unexpected since CK1 and PKC are Ser/Thr protein kinases in higher eukaryotes. Whether TcCK1 and TcPKC1 are dual protein kinases in *T. cruzi* remains to be determined. However, support for this hypothesis came from the fact that TcCK1 is in vitro autophosphorylated on Tyr residues, as indicated by Western blot analysis (this work). Further studies will be necessary to determine whether TcCK1 is in vivo autophosphorylated. TcPKC1 also gets autophosphorylated when a radioactive label is used [39]; however, we did not detect TcPKC1 autophosphorylation using Western blot analysis (this work), presumably due to the low phosphorylation level.

Notably, TcCK1, TcCK2, TcAUK1, and TcPKC1 can in vitro phosphorylate identical Ser and Thr residues on TcPolβ, since all four protein kinases phosphorylate Ser69 and Ser275, while TcCK2, TcCK1, and TcPKC1 phosphorylate Thr382. Also, TcCK1 and TcPKC1 phosphorylate identical Ser residues, namely Ser13, Ser46, Ser69, Ser185, and Ser275. On the other hand, TcCK1, TcCK2, and TcPKC1 can phosphorylate Thr382 on TcPolβ, while TcCK1 and TcPKC1 phosphorylate Thr49. This was rather unexpected since eukaryotic protein kinases can modify kinase-specific phosphorylation sites on their protein substrates [50]. However, *T. cruzi* is divergent from higher eukaryotes, and they have evolved unique mechanisms and strategies to regulate their complex life cycle and to adapt to different hosts. In this regard, it might be possible that their protein kinases could have slightly different specificities compared with their higher eukaryote counterparts. Moreover, protein kinase specificities have not been studied in trypanosomatids, and their recognition mechanisms are largely unknown. In the cellular context, protein kinase activities are highly and tightly regulated by several mechanisms including sequence motifs, subcellular location, translocation, post-translational modifications, protein–protein interactions, regulatory subunits, dimerization, allosteric interactions with activators or inhibitors, and autoinhibitory domains, among others [51,52]. In a recent study, it has been described that several distantly related kinases, in the YANK, CK1, CK2, GRK, and TGFβ receptor families, converged to phosphorylate similar sequence motifs despite their different locations on the kinome tree [53]. Nevertheless, our studies indicate that TcPolβ is in vitro phosphorylated by TcCK1, TcCK2, TcPKC1, and TcAUK1, which increases the DNA synthesis activity of this Ser/Thr-rich protein substrate.

It is worth mentioning that rat DNA Pol β is in vitro phosphorylated by PKC, which leads to the inactivation of the DNA polymerase beta activity [54]. PKC phosphorylates the rat DNA Pol β at two Ser residues, namely Ser44 and 53, at the N-terminus lyase domain of this DNA polymerase. Recently, it has been demonstrated that phosphorylation of Ser44 causes a conformational change in the structure of DNA Pol β, causing a transition from a closed to an open structure [55]. However, in vivo, mammalian DNA Pol β phosphorylation by PKC has not yet been demonstrated, nor by another protein kinase. The in vitro results suggest that phosphorylation could regulate mammalian DNA Pol β activity in vivo.

It has been found in a phosphoproteomic study in *T. cruzi* that phosphorylated proteins (753) containing 2572phosphorylation sites have multiple phosphorylation sites, with 73% containing more than two and 53% containing three or more phosphorylation sites. The Ser/Thr/Tyr distribution of the phosphorylated sites which were identified mapped to 84.1% pSer, 14.9% pThr, and 1.0% pTyr amino acid residues and several protein kinases were found to be phosphorylated on Tyr residues, for example, PKC [56]. Thus, more than 30% of the phosphorylated Tyr residues detected have been found on protein kinases. Therefore, this means that *T. cruzi* kinome has a Ser/Thr/Tyr phosphorylation site distribution shifted toward pTyr, with 6% of all the phosphorylation sites found on protein kinases being pTyr, displaying a 6-fold enrichment compared to the rate found for the total phosphoproteome [56]. Since the *T. cruzi* genome does not encode typical Tyr protein kinases, it has been proposed that Tyr phosphorylation in trypanosomatids is probably due to the activity of atypical Tyr protein kinases as WEE1-like protein kinases and dual Tyr phosphorylation-regulated protein kinases that can phosphorylate Ser/Thr and Tyr residues as well [57]. Indeed, the trypanosomatids possess a large set of protein kinases, which is close to 2% of each genome, suggesting a key role of protein phosphorylation in the lives of trypanosomatid parasites, relying on downstream molecules that perform stage and cell cycle-specific functions.

Another striking observation in this work is that most of the phosphorylated peptides of TcPolβ by the analyzed *T. cruzi* protein kinases do not match the consensus recognition sites described for higher (eukaryotic) protein kinases, for example, CK2 phosphorylates Ser/Thr residues in an acidic context. TcPolβ has several CK2 consensus sites at the C-terminal region; however, these were not phosphorylated by TcCK2, raising the possibility that *T. cruzi* protein kinases may have different specificities compared with their mammalian counterparts. On the other hand, the in vivo phosphorylation sites found in our earlier report were different from the in vitro phosphorylation by TcCK2 and TcPKC1, which suggests that other protein kinases might in vivo phosphorylate these sites [39]. Perhaps, as in the case of PKC, another isoenzyme (TcPKC2 or TcPKC3) might in vivo phosphorylate those sites.

The phorbol ester family, DAG, Ca^+2^, and PS directly activate PKC in higher eukaryotes [58]. PMA can induce morphological changes in filopodium-like structures from the flagellar membrane in *T. cruzi* epimastigotes. Also, PMA can significantly increase the attachment and ingestion of epimastigotes by resident or activated mouse peritoneal macrophages. Strikingly, PMA does not affect trypomastigotes [59]. Since PMA binds to membrane-associated PKC, these results could be attributed to PKC signaling. PKC activity has been characterized from *T. cruzi* epimastigotes cell extracts and was found in membrane and solubilized membrane fractions [45,46]. Our results show that TcPKC1 can be in vitro activated by PS as measured by Western blot using TcPolβ as substrate. Also, PMA can stimulate TcPK1, as detected by Western blot analysis and by a functional assay, in which PMA can in vitro activate TcPKC1 to phosphorylate TcPolβ, augmenting its DNA synthesis activity. TcPKC1 is predicted to have a PhoX domain at the N-terminus, which could bind phosphoinositol phosphates (PIPs) [39]. Also, it has a lipocalin domain at the C-terminus that can bind free fatty acids. The activators most likely can bind to these domains and stimulate their phosphorylation activity [39]. PMA can also stimulate in vivo TcPolβ phosphorylation in the proliferative epimastigote forms; however, we could not observe a similar stimulation in the non-proliferative trypomastigote forms under the same conditions. The phosphorylation stimulation on TcPolβ must be the direct effect of PMA on TcPKC since this phorbol ester is a stimulator of PKC activity in higher eukaryotes. This result suggests that *T. cruzi* epimastigotes, but not tissue-cultured trypomastigotes, express PKC in an active form.

Finally, the molecular studies of protein kinases and DNA polymerases in trypanosomatids are worth pursuing. DNA polymerases are essential for replication and DNA repair. On the other hand, protein kinases are important for cell differentiation, proliferation, and stress response during the complex life cycle of trypanosomatids. Protein kinases represent attractive and promising drug targets for diseases caused by trypanosomatids. Despite some homology with their mammalian counterparts, trypanosomatid protein kinases are significantly different in such a way that opens up the possibility of developing parasite-selective protein kinase inhibitors.

## Figures and Tables

**Figure 1 microorganisms-12-00907-f001:**
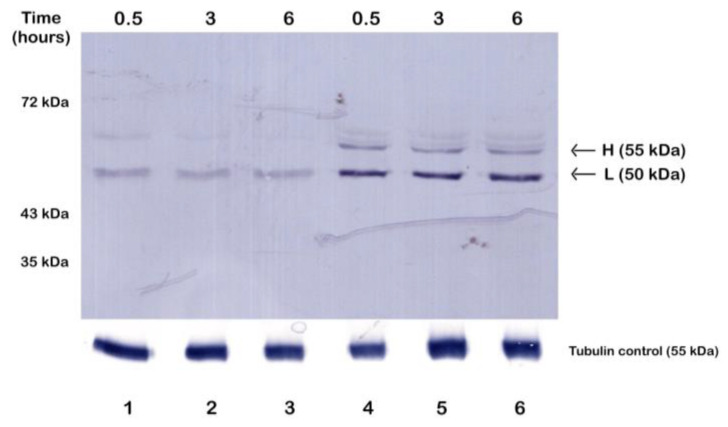
Effect of Fetal Calf Serum on TcPolβ protein synthesis in *T. cruzi* trypomastigotes. Tissue-cultured trypomastigotes were incubated in RPMI1640 medium without heat-inactivated FCS (lanes 1–3) or in RPMI1640 supplemented with 5% *v*/*v* of heat-inactivated FCS (lanes 4–6) and the levels of TcPolβ were measured by Western blot at 0.5, 3, and 6 h post incubation (upper panel). Each lane contains the equivalent of 250,000 parasites (approximately 4 µg total protein). The lower panel shows a Western blot of tubulin as a control. Parasites were mobile and viable at 6 h post incubation.

**Figure 2 microorganisms-12-00907-f002:**
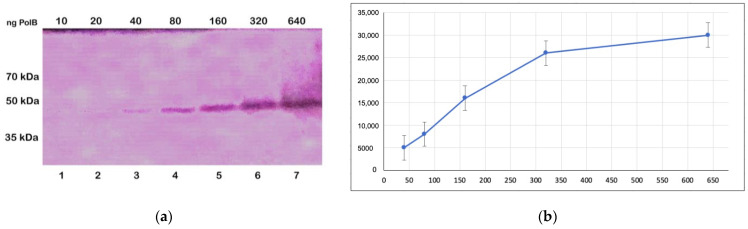
DNA polymerase gel activity assay followed by colorimetric detection. (**a**) Different amounts of recombinant TcPolβ (10–640 ng) were separated on a 10% PAGE-SDS, renatured, and incubated with a solution containing dNTPs, including dATP-16-Biotin labeled and detected using Streptavidin-Alkaline phosphatase as described in the Section 2. The assay’s detection limit is 40 ng, and a linear dose–response was obtained with this technique. (**b**) A plot was obtained from the quantification of Figure 2a.

**Figure 3 microorganisms-12-00907-f003:**
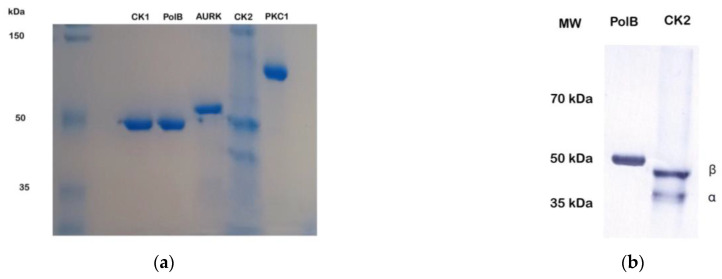
(**a**) PAGE-SDS gel of purified recombinant proteins, which were used in this work. The figure shows a 10% PAGE-SDS stained with colloidal Coomassie Blue G-250. One hundred nanograms (100 ng) of the different proteins, indicated at the top of the figure, were loaded and analyzed for PAGE-SDS. At the left side of the figure, protein MW markers are shown. (**b**) Western blot of the CK2 (αβ) holoenzyme with anti-His tag monoclonal antibodies. Both subunits are not degraded as can be seen from the figure.

**Figure 4 microorganisms-12-00907-f004:**
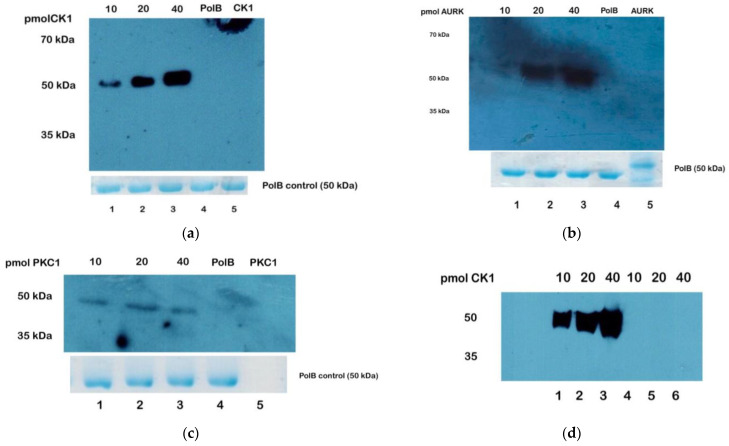
Western blot analysis of TcPolβ phosphorylation by different *T. cruzi* protein kinases and detected by anti-phosphor antibodies. Four hundred nanograms (400 ng) of TcPolβ were incubated with 1 mM ATP and different amounts of the protein kinases (10, 20, and 40 pmol of each), as indicated at the top of the figure, and subjected to PAGE-SDS, followed by Western analysis. The unphosphorylated TcPolβ (400 ng) and the analyzed protein kinase (40 pmol) were used as a control. (**a**) Phosphorylation of TcPolβ by CK1 protein kinase. (**b**) Phosphorylation by AUK1 (AURK) protein kinase. (**c**) Phosphorylation by PKC1 protein kinase. Western blots of figures (**a**–**c**) were analyzed with anti-phosphor Ser-Thr-Tyr monoclonal antibodies. Polβ loading control is in the lower panel of figures (**a**–**c**). (**d**) Autophosphorylation of CK1 on Tyr residues. Different amounts of CK1 (10, 20, and 40 pmol) were incubated with 1 mM ATP. As a control, the same amounts of CK1 were incubated in the absence of ATP. Afterward, Western blot analysis analyzed the reactions with specific anti-phosphor Tyr monoclonal antibodies.

**Figure 5 microorganisms-12-00907-f005:**
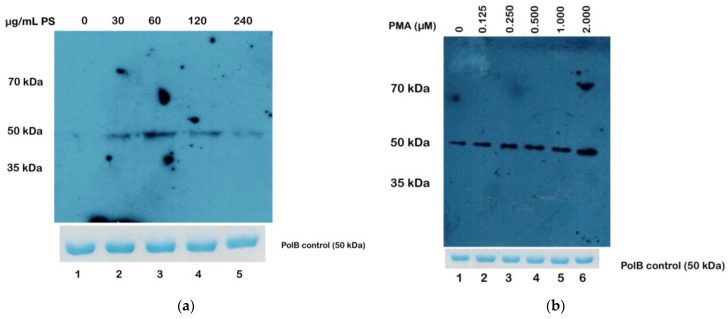
Phosphatidylserine and PMA stimulate PKC1 phosphorylation activity. (**a**). One hundred nanograms (100 ng) of TcPolβ were incubated with 1 mM ATP and different amounts of PS (30–240 µg/mL), as indicated at the top of the figure, and analyzed by Western blot using anti-phosphor Ser-Thr-Tyr monoclonal antibodies. (**b**) One hundred nanograms (100 ng) of TcPolβ were incubated with 1 mM ATP and different amounts of PMA (0.125–2.00 µM) as indicated at the top of the figure and analyzed by Western blot with the same antibodies used in Figure 4a. Lane 1 contains only TcPolβ plus DMSO. In both figures, a Polβ loading control was included in the lower panel from a Coomassie Blue G-250 stained PAGE-SDS gel.

**Figure 6 microorganisms-12-00907-f006:**
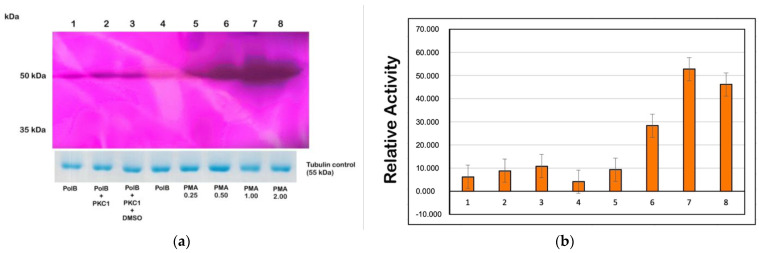
PMA stimulates PKC1 to augment the DNA synthesis activity of TcPolβ. The PMA effect was measured by a DNA polymerase gel activity assay followed by colorimetric detection. (**a**) Forty nanograms (40 ng) of TcPolβ were used in the experiment in all lanes from 1 to 8 and incubated under several conditions. Lane 1 contains only TcPolβ, while lane 2 contains 40 pmol of PKC1 plus 1 mM ATP. Lane 3 contains the same components as lane 2 plus DMSO. Lane 4 contains TcPolβ and 1 mM ATP. Lanes 5–8 contain TcPolβ, 40 pmol of PKC1, 1 mM ATP, and increasing amounts of PMA from 0.25–2.00 µM as indicated at the bottom of the figure. (**b**) Quantification of (**a**). A Polβ loading control from a Coomassie Blue G-250 stained PAGE-SDS was included, as seen at the bottom of Figure 3a.

**Figure 7 microorganisms-12-00907-f007:**
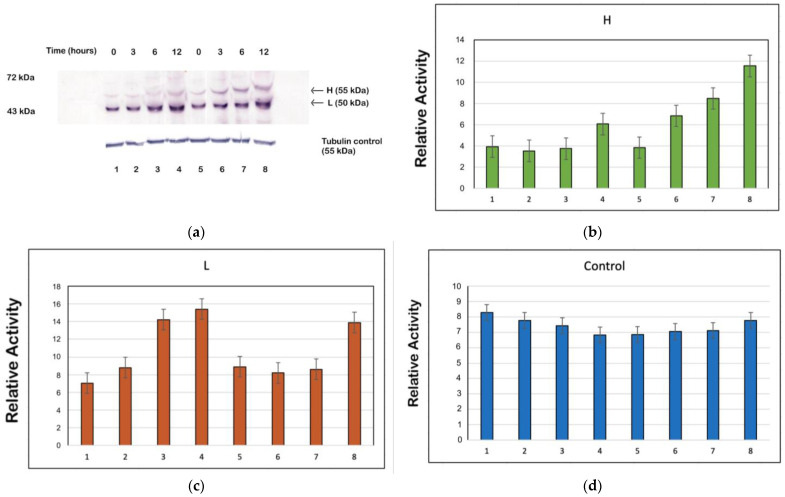
PMA stimulates in vivo PKC to increase the phosphorylation of TcPolβ in *T. cruzi* epimastigotes. (**a**) Epimastigote cells were incubated with 2 µM of PMA, and samples were taken at 0, 3, 6, and 12 h and analyzed by Western blot with anti-TcPolβ antibodies (upper panel, lanes 5–8). Controls (lanes 1–4) contained only DMSO. All lanes contained total protein from the equivalent of 250,000 parasites (roughly 4 µg total protein). Tubulin was used as a control (lower panel). (**b**) Quantification of the H form of TcPolβ from (**a**). (**c**) Quantification of the L form of TcPolβ from (**a**). (**d**) Quantification of the tubulin control from (**a**).

**Table 1 microorganisms-12-00907-t001:** Phosphorylated residues on TcPolβ by the indicated protein kinases as determined by MS analysis.

Kinase	Phosphorylated Residues on TcPolβ			
AUK1	S 69	S 275	T 366	
CK2	S 69	S 275	T 382	
CK1	S 13	S 46	T 49	S 69
	S 185	S 193	S 275	T 278
	Y 307	S 336	T 338	T 345
	T 382			
PKC1	S 13	S 46	T 49	S 69
	S 185	S 275	T 382	Y 302

## Data Availability

Data are contained within the article and Appendix A.

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
