# Peer review of "In Vitro Identification of Phosphorylation Sites on TcPolβ by Protein Kinases TcCK1, TcCK2, TcAUK1, and TcPKC1 and Effect of Phorbol Ester on Activation by TcPKC of TcPolβ in Trypanosoma cruzi Epimastigotes"

_microorganisms, 2024, doi:10.3390/microorganisms12050907_

Round 1

Reviewer 1 Report

Comments and Suggestions for Authors

The reviewed article presents a modern analysis (using the latest research methods) of mechanisms related to the development and pathogenic effects of the particularly important pathogen - Trypanosoma crusi.

Due to globalization and specific pathogenesis, it is a parasite that threatens a large population of people in different places and times. Apart from the cognitive nature of the research described, they are the basis for searching for new preventive and therapeutic solutions. The work is very interesting and important for the development of medicine. It presents high professionalism of the research contractors. I do not make any substantive comments regarding the content of the article.

  However, I have suggestions regarding editorial aspects.

  In my opinion, the authors expanded the introduction part too much. There is very detailed information correlating with the scope of research, which could be included in the discussion. There is also a missing part related to the research objectives. Here, the authors presented information related to the scope of the research performed. To achieve a better message, I suggest presenting the assumed research goals in bullet points, and in the discussion a reference to their achievement.

  After taking into account these minor corrections, the work is ready for publication.

Reviewer 2 Report

Comments and Suggestions for Authors

Dear Authors,

The manuscript is very interesting for the scientific community, and the results look promising. It could be a good article.

However, it needs significant improvement. Some observations that need to be clarified are in the lanes below.

Lane 173. Add IPTG concentration information.

Lane 186. Instead of ul, could the authors add concentration information?

Lane 188, 100 ug/ml of activated calf thymus DNA; does it mean 2ug? (since the authors loaded 20 ul). Please describe the materials and methods more precisely and accurately.

Lane 189. Please describe the materials used in the loading buffer (including concentration information).

Lane 218. Electrophoresis does not fractionate; it is a protein separation based on molecular weight. Please remove or clarify the sentence.

Lane 236. Is the FCS heat inactivated? Lanes 156 and 158 generate confusion, and clarification is needed.

Lane 245. Add information about the number of trypomastigotes or protein concentration.

Lanes 247 and 248. The authors did not use anti-phospho antibodies (at least is not described in the legend), so the following sentence is wrong: “Those results indicate that FCS components signal β protein synthesis and phosphorylation in trypomastigotes”.

We must show evidence before concluding.

Lane 254. Do the 250,000 parasites have less than 50ug of total protein? 

Protein concentrations for electrophoresis and WB usually are between 20-50 ug (total), or 0.2-2ug (recombinant protein); please clarify.

Figure 1. A molecular weight ladder for both WB results is needed.

Also, information about the molecular weight of H and L forms should be added. 

Tub is Tubuline? It must be described somewhere, and the molecular weight of the tubulin protein must be added.

Lane 264. Add the information about results obtained for DNA polymerase Klenow in supplementary materials.

Lane 267. Again, we can only conclude with data. Please add the information on how the fold times were estimated (any statistical method, including information about experimental and biological replicates).

Figure 2A. Add molecular weight marker and the respective molecular weight for the protein separated by electrophoresis.

Figure 2B. Does not show any standard deviation information. Please always work following scientific methodology (triplicates biological and experimental).

We should always use a duplicate membrane or proteins pre-labeled for the molecular weight ladder (for example, dual color, etc).

Lane 301: Is 50 ng of protein for Figure 3? It should be an error. Coomassie Blue G-250 cannot detect that low an amount of protein; see the marker; it barely detects the proteins in the ladder.

Also, CK2 protein is degraded; did the authors have another gel? (maybe from the replicates)

Again, information about the molecular weight of each protein studied should be added.

Which photo doc equipment was used to obtain Figure 3? This information should be included in the manuscript.

Based on the previous errors, we need the raw data to see if the information presented in Table 1 is correct; please add the raw data to a supplementary file.

If Figure 3 is the only one used for MS/MS?, then all “degraded” CK2 discussions could lead to misleading information.

Figure 4. The molecular weight ladder needs to be included, and we also need a membrane duplicate to see all membranes (Coomassie staining or any other staining methodology). Those films are small pieces (cut), and we cannot see the specificity of the antibodies.

Figure 4. The legend must indicate which antibody was used for each membrane, which generates confusion. Please rephrase the legend and describe it more accurately.

Figure 5. Ladder?

In 5a and 5b, the authors described using 100 ng for each protein; the results do not show an equal protein concentration. We need a duplicate membrane to confirm this. Please describe how the protein concentration was quantified. Is it a Bradford or another more sensitive strategy/methodology?

Figure 6. 40 ng? It does not look like a low amount of protein is loaded, especially if we see the reactions on lanes 5-8. Something needs to be more accurate.

Figure 6b. We need information about experimental and biological replicates, then add information about the deviation standard, statistical analysis, etc. The same observation is for Figures 7 b-d.

Figure 7a, molecular weight ladder? Information about the molecular weight of the protein of interest needs to be included.
